# Seeing Down the Line: Endoscopic Reconstruction with Centerline Constraints

**Andrea Dunn Beltran**[1]                                        ASDUNNBE@CS.UNC.EDU
[1] *The University of North Carolina, Department of Computer Science, Chapel Hill, NC, USA*

**Romain Hardy**[2]                                         ROMAIN_HARDY@FAS.HARVARD.EDU
**Pranav Rajpurkar**[2]                              PRANAV_RAJPURKAR@HMS.HARVARD.EDU
[2] *Harvard University, Department of Biomedical Informatics, Boston, MA, USA*

**Editors:** Accepted for publication at MIDL 2026

## Abstract

Colonoscopy remains the gold standard for colorectal cancer screening, but there is still no real-time, geometry-aware way to quantify which parts of the colon have been inspected during a procedure. We revisit 3D Gaussian endoscopic reconstruction as a representation and geometry problem rather than a new network design. Assuming known camera poses and off-the-shelf depth or photometric supervision, we add a simple centerline-based coordinate system and priors on top of an existing Gaussian mapping backbone. From the noisy pose stream we maintain an online centerline and Bishop frame, assign each Gaussian tubular coordinates $(s, r, \theta)$, and use these coordinates both to regularize the map toward a hollow tube and to accumulate coverage statistics in colon-intrinsic space. On long C3VD phantom colonoscopy sequences, this lightweight modification achieves Chamfer distance comparable to or better than an endoscopy-specific 3D Gaussian SLAM baseline while running at frame rates close to MonoGS and yielding improved rendering quality, with negligible additional computation. At the same time, the same representation produces unrolled colon views and segment-wise coverage summaries essentially "for free", making centerline-aware Gaussian mapping a practical drop-in component for future real-time quality monitoring tools in colonoscopy.

**Keywords:** Endoscopy, 3D Reconstruction, Coverage

## 1. Introduction

Colorectal cancer is a leading cause of cancer-related death, yet it is highly preventable when precancerous polyps are detected and removed early (Siegel et al., 2024). Colonoscopy is the gold standard for screening and therapy (Rex et al., 2015), but tandem studies still report substantial adenoma miss rates, especially for flat and serrated lesions hidden behind folds (Zhao et al., 2019). Current quality indicators such as adenoma detection rate and withdrawal time have improved practice (Kaminski et al., 2010), but they are coarse and retrospective: they do not say which parts of the colon were actually inspected and how well during a particular procedure.

At the same time, 3D reconstruction methods for endoscopy have matured. Recent 3D Gaussian SLAM systems can build detailed maps from monocular endoscopic video (Matsuki et al., 2024; Wang et al., 2024), and they are increasingly considered as building blocks for navigation and documentation. However, most systems are optimized either for general scenes or for short surgical clips, and treat the reconstruction as an end in itself. They do

not explicitly encode colon anatomy or provide real-time coverage summaries that match how endoscopists report procedures (segment, insertion depth, circumferential position). In practice, the cost of adding such structure is a concern: clinical systems will favor small, robust changes over entirely new networks.

We revisit endoscopic 3D Gaussian mapping from this angle. Assuming that camera poses are provided by an external tracker or SLAM system and that depth or photometric supervision is available, we ask: *How much benefit can we obtain from a minimal geometric modification to an existing system?* Our answer is to keep the backbone unchanged and add a simple centerline-based coordinate system and priors. From the noisy pose stream we maintain an online colon centerline and Bishop frame, assign each Gaussian tubular coordinates $(s, r, \theta)$, and use these coordinates to (i) regularize the Gaussians toward a hollow tube around the centerline and (ii) accumulate online coverage statistics in colon-intrinsic space. This requires some additional geometry and bookkeeping, but only modest extra computation.

On long C3VD phantom colonoscopy sequences (Bobrow et al., 2023), this lightweight modification yields a useful trade-off: it matches or improves the Chamfer distance of EndoGSLAM (Wang et al., 2024), runs at effective frame rates close to a MonoGS-style mapper (Matsuki et al., 2024), and provides better held-out renderings, while simultaneously producing unrolled colon views and segment-wise coverage summaries "for free." We do not claim a fundamentally new representation or learning paradigm; rather, we show that making colon anatomy a first-class constraint in an otherwise standard Gaussian mapping pipeline gives clinically relevant outputs at minimal incremental cost.

Our contributions are:

- **Centerline-aware Gaussian mapping.** We extend a standard 3D Gaussian mapper with an online colon centerline and Bishop frame, assigning each Gaussian tubular coordinates without changing the underlying rendering or optimization machinery.
- **Cheap colon-specific priors and coverage.** Using these coordinates, we add simple tube and smoothness priors that keep Gaussians near the mucosal surface and accumulate per-segment coverage statistics in real time, yielding unrolled coverage maps essentially for free.
- **Evaluation on long, physician recorded sequences.** On C3VD, our method achieves EndoGSLAM level Chamfer distance with MonoGS-like frame rates and better rendering quality, while providing online coverage maps and segment summaries not available from either baseline.

## 2. Related work

**Per-frame assistance and quality assessment.** Deep learning has enabled real-time computer-aided detection and diagnosis in colonoscopy, with systems that highlight polyps, classify lesion types, and estimate per-frame quality scores (Urban et al., 2018; Wang et al., 2019). Large datasets such as HyperKvasir (Borgli et al., 2020) have supported multi-class lesion detection and automated quality assessment, including withdrawal speed and bowel preparation (Chang et al., 2022). These approaches operate mainly in image space, reasoning over individual frames or short clips. They do not maintain a persistent 3D representation of the colon or provide explicit, geometry-based coverage measures.

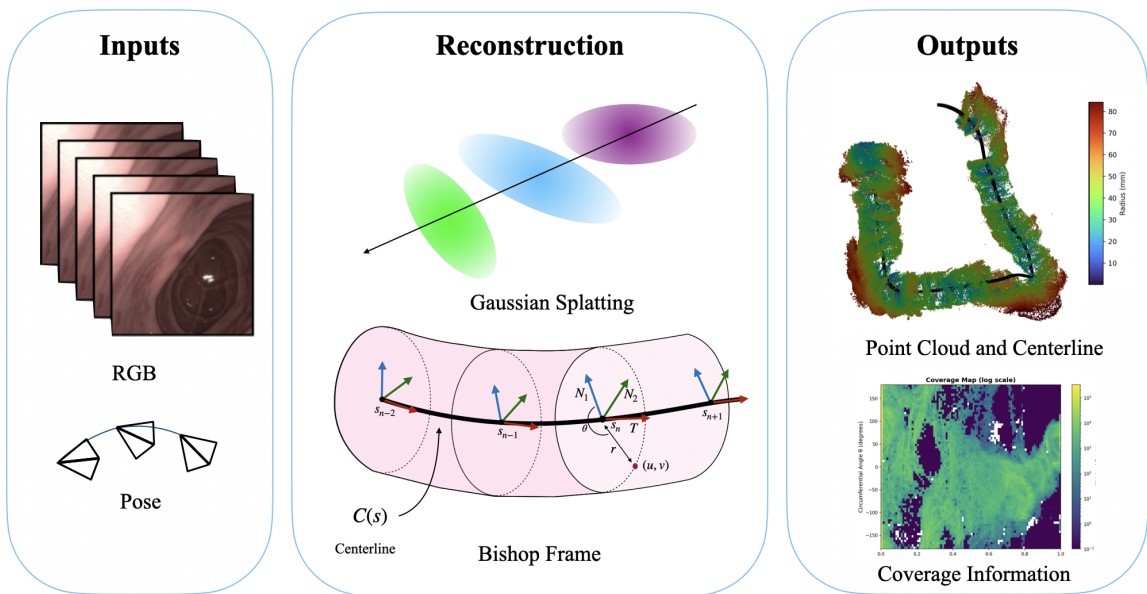

Figure 1: Overview of our centerline-aware 3D Gaussian mapping pipeline. **Inputs:** monocular colonoscopy RGB frames and externally provided poses. **Reconstruction:** a thin geometric layer around a standard 3D Gaussian mapper maintains an online centerline $C(s)$ and Bishop frame, assigns tubular coordinates $(s, r, \theta)$, and updates keyframes and coverage counters in colon-intrinsic space. **Outputs:** a 3D Gaussian reconstruction with centerline and online coverage maps in $(s, \theta)$, providing segment-wise coverage summaries with minimal extra computation.

**Dense SLAM and endoscopic reconstruction.** Dense SLAM has evolved from volumetric fusion to neural implicit and 3D Gaussian representations (Zhu et al., 2022; Matsuki et al., 2024; Keetha et al., 2024). Endoscopic variants adapt these ideas to deformable and specular anatomy. RNNSLAM couples a recurrent depth-and-pose network with a SLAM backend for colon reconstruction (Ma et al., 2021), while EndoGSLAM integrates 3D Gaussian splatting into endoscopic surgery, demonstrating real-time tracking and dense mapping (Wang et al., 2024). The C3VD dataset provides phantom colonoscopy videos with depth and ground-truth geometry, enabling quantitative evaluation (Bobrow et al., 2023). These systems focus on geometry and tracking accuracy; they typically treat the colon as a generic scene and do not organize the map in colon-intrinsic coordinates or expose coverage metrics as first-class outputs. Our work builds directly on this line, but holds pose fixed and instead modifies the representation and loss to be colon-aware.

**Tubular and anatomical priors.** Tubular priors are widely used to model elongated anatomical structures such as vessels and airways (Bauer and Bischof, 2009; Chlebiej et al., 2023). For the colon, prior work has explored digital "unrolling" for visualization and registration (Rossides et al., 2021) and tubular non-rigid structure-from-motion models for colonoscopic video (Sengupta et al., 2021; Floor et al., 2022). These methods show the value of encoding tube-like anatomy, but are typically offline and not framed as small

modifications to existing real-time reconstruction pipelines. In contrast, we integrate a simple tubular coordinate system and associated priors directly into a Gaussian mapper, and show that this modest geometric addition is enough to match strong baselines in geometry while providing online coverage information at negligible extra cost.

## 3. Method

Our only modification to a standard 3D Gaussian mapper is to wrap it in a colon-intrinsic coordinate system based on an online centerline and its Bishop frame (Figure 1). This provides a natural parameter $s$ for insertion depth, tubular coordinates $(s, r, \theta)$ for each Gaussian, and cheap coverage statistics in the same space. All rendering, networks, and optimizers are unchanged from a MonoGS-style backbone; implementation details and hyperparameters are given in Appendix A. Throughout, we treat camera poses and depth as upstream inputs and focus on the mapping/representation layer; correcting slow pose drift or improving depth prediction is beyond the scope of this work. Our key intuition is that this thin geometric module is sufficient to unlock both better use of Gaussian capacity and coverage-aware outputs at minimal additional cost.

### 3.1. Gaussian map and reconstruction loss

Let $\{T_t\}_{t=1}^N \subset SE(3)$ denote per-frame camera poses and $\{I_t\}_{t=1}^N$ the corresponding RGB images. We maintain a set of 3D Gaussian primitives $\{g_i\}_{i=1}^M$, where each $g_i$ has mean $\mu_i \in \mathbb{R}^3$, anisotropic covariance $S_i$, color, and opacity. A differentiable Gaussian rasterizer projects $\{g_i\}$ into keyframe views $T_k$ to produce rendered images $\hat{I}_k$ and, where available, depth maps $\hat{D}_k$. Using observed keyframe RGB $I_k$ and depth $D_k$, we adopt standard photometric and depth reconstruction losses

$$\mathcal{L}_{\text{photo}} = \sum_{k \in \mathcal{K}} \sum_{p \in \Omega_k} \left\| I_k(p) - \hat{I}_k(p) \right\|_1, \qquad \mathcal{L}_{\text{depth}} = \sum_{k \in \mathcal{K}} \sum_{p \in \Omega_k} \left\| D_k(p) - \hat{D}_k(p) \right\|_1, \qquad (1)$$

where $\mathcal{K}$ is the set of keyframes and $\Omega_k$ the valid pixels. We do not introduce new networks or detectors; all supervision comes from existing tracking and depth components.

**Front-end / back-end separation.** The system is split into a light *front-end* and a heavier *back-end*:
- **Front-end.** Processes every frame, maintains an online centerline $C(s)$, its Bishop frame, an arc-length-based keyframe schedule, and low-cost coverage counters in colon coordinates. This runs at the native video frame rate and can provide insertion depth and coarse coverage in real time.
- **Back-end.** Operates only on keyframes. Given the current centerline, it assigns tubular coordinates to Gaussians, evaluates colon priors, and updates the Gaussian map using the combined loss. This mirrors the mapping component of MonoGS-style 3D Gaussian reconstruction, but treats poses as fixed and injects colon-aware geometric constraints.

This separation keeps the extra computation marginal while allowing the Gaussian optimizer to focus on fewer, well-chosen views.

### 3.2. Online centerline and Bishop frame

Our goal is to convert noisy colonoscopy motion into a smooth, 1D backbone that approximates the scope path and serves as a clinically meaningful coordinate for insertion depth and segment location.

**Backbone construction.** We represent the exploration path by a centerline curve $C(s) \in \mathbb{R}^3$ parameterized by arc length $s$. Online, we maintain a sparse set of *backbone points* $\{b_k\}_{k=1}^K$ derived from camera centers $x_t$. A new backbone point is added when (i) the translational motion since $b_K$ exceeds $d_{\min}$, (ii) the motion direction does not exceed a maximum bend angle, and (iii) the candidate is not within $d_{\text{loop}}$ of non-neighbor backbone points (loop avoidance). This filters small jitter while following the exploration path; full thresholds are listed in Appendix A.

**B-spline smoothing and arc length.** Given $\{b_k\}$, we fit a cubic B-spline $\tilde{C}(u)$, $u \in [0, 1]$, and sample it at $\{u_m\}_{m=0}^M$. We compute cumulative distances $s_0 = 0$, $s_m = s_{m-1} + \|\tilde{C}(u_m) - \tilde{C}(u_{m-1})\|_2$, store positions $C_m = \tilde{C}(u_m)$ and arc-lengths $s_m$, and query intermediate points by linear interpolation in $s$. The coordinate $s$ is therefore a smoothed approximation of physical insertion depth, more useful for documentation and coverage than frame index or raw Euclidean distance. We found that this strategy reduces sensitivity to high-frequency pose jitter, as seen in Figure 5 in Appendix F, but it does not correct slow systematic drift.

**Bishop frame.** To define a stable tubular coordinate system we compute a Bishop frame $\{T(s), N_1(s), N_2(s)\}$ along $C$. For each sample $C_m$ we estimate the tangent $T_m$ by finite differences and propagate an orthonormal pair of normals $(N_{1,m}, N_{2,m})$ using discrete parallel transport, i.e. the minimal rotation taking $T_{m-1}$ to $T_m$ followed by re-orthogonalization. This yields an orthonormal basis with minimal twist at each $s_m$, and is numerically stable even in nearly straight segments where a Frenet frame would be ill-conditioned.

We initialize $N_{1,0}$ by projecting a fixed reference direction (e.g. the first keyframe camera "up" vector) onto the plane orthogonal to $T_0$ and normalizing; we set $N_{2,0} = T_0 \times N_{1,0}$. To prevent occasional sign flips from discretization/noise, we enforce continuity by checking $\langle N_{1,m}, N_{1,m-1} \rangle$: if negative, we flip both normals $(N_{1,m}, N_{2,m}) \leftarrow (-N_{1,m}, -N_{2,m})$. This preserves a consistent $\theta$ convention for unrolled coverage.

### 3.3. Colon-intrinsic coordinates

The Bishop frame turns the colon into a tubular coordinate system. Given a 3D point $x$ (e.g. a Gaussian center $\mu_i$), we first find its closest point on the centerline by minimizing $\|x - C(s)\|_2$ over sampled $\{s_m\}$ and, optionally, refining with a 1D line search, obtaining $s^\star$ and $C^\star = C(s^\star)$. We then query the Bishop frame at $s^\star$, $T^\star, N_1^\star, N_2^\star$, and express the offset $\Delta x = x - C^\star$ as

$$u = \Delta x \cdot N_1^\star, \quad v = \Delta x \cdot N_2^\star, \quad \ell = \Delta x \cdot T^\star.$$

The radial distance and circumferential angle are

$$r(x) = \sqrt{u^2 + v^2}, \qquad \theta(x) = \text{atan2}(v, u),$$

so that $(s^\star, r(x), \theta(x))$ defines colon-intrinsic coordinates for $x$. We use these coordinates for keyframe selection, colon-aware priors, and coverage accumulation. Clinically, $s^\star$ aligns with insertion depth and segment, while $\theta(x)$ captures circumferential location on the mucosal surface. We define $\theta = 0$ along $N_1(s)$ and increase $\theta$ toward $N_2(s)$ via the right-hand rule around $T(s)$. For binning and visualization, we wrap $\theta$ to $(-\pi, \pi]$ (or equivalently $[0, 2\pi)$) with circular boundary handling.

### 3.4. Arc-length-based keyframing

Standard keyframe selection thresholds on Euclidean camera motion or image overlap, which oversamples straight segments and undersamples tight bends in a tubular organ. We instead select keyframes by distance traveled along the centerline.

At frame $t$ with camera center $x_t$, we estimate the arc-length increment $\Delta s_t$ since frame $t-1$: if the centerline already covers both positions, we project $x_{t-1}$ and $x_t$ onto $C$ to obtain $s_{t-1}$ and $s_t$ and set $\Delta s_t = |s_t - s_{t-1}|$; otherwise, while the centerline is still growing, we approximate forward progress using the tangent at the endpoint and the signed projection of $(x_t - x_{t-1})$ onto that tangent, clipped at zero. We maintain an accumulator $A_t = A_{t-1} + \Delta s_t$ with $A_0 = 0$ and create a new keyframe when standard appearance- and time-based criteria are met and $A_t \geq \tau_{\text{KF}}$, where $\tau_{\text{KF}}$ is the desired spacing in millimeters. The accumulator is then reset. This yields approximately uniform keyframe density in $s$ and automatically densifies keyframes in high-curvature segments where coverage is more challenging.

### 3.5. Colon-aware priors and total loss

Once tubular coordinates are defined, we use them to regularize the reconstruction toward an anatomically plausible hollow tube with a smooth backbone. This ties the Gaussian map to colon geometry and reduces the search space for the optimizer.

**Radial tube prior.** The colon wall is approximately tubular around the centerline at a characteristic radius $r_{\text{wall}}$. For each Gaussian $i$ with center $x_i$ we compute its radius $r_i = r(x_i)$ and penalize deviations from $R_{\text{wall}}$ via

$$\mathcal{L}_{\text{radial}} = \sum_i \rho\big((r_i - r_{\text{wall}})^2\big), \tag{2}$$

where $\rho(\cdot)$ is a robust penalty. This term is a soft, robust regularizer: it primarily discourages lumen "fill-in" and off-wall outliers while the data terms determine local wall shape.

**Centerline smoothness prior.** To avoid spurious kinks from noisy motion, we regularize the discrete second derivative of the sampled centerline positions $\{C_m\}$:

$$\mathcal{L}_{\text{curv}} = \sum_{m=1}^{M-1} \big\|C_{m+1} - 2C_m + C_{m-1}\big\|_2^2. \tag{3}$$

This penalizes rapid curvature changes while allowing anatomically plausible bending and yields a reliable backbone for reporting and unrolling.

**Total loss.** The back-end minimizes a combination of photometric, geometric, and colon-aware terms:

$$\mathcal{L} = \mathcal{L}_{\mathrm{photo}} + \mathcal{L}_{\mathrm{depth}} + \lambda_{\mathrm{radial}}\,\mathcal{L}_{\mathrm{radial}} + \lambda_{\mathrm{curv}}\,\mathcal{L}_{\mathrm{curv}}, \tag{4}$$

with scalar weights $\lambda_{\mathrm{radial}}$ and $\lambda_{\mathrm{curv}}$. The same tubular coordinates used to define these priors are also used to compute unrolled coverage maps and segment-wise coverage statistics in Section 4, so the representation that improves geometry and efficiency also directly exposes clinically meaningful readouts.

## 4. Experiments

We evaluate whether a thin layer of colon-aware geometry on top of a standard 3D Gaussian mapper can (i) match the geometric accuracy of an endoscopy-specific 3DGS baseline, (ii) retain the frame rates of a MonoGS-style mapper, and (iii) expose useful online coverage information at negligible additional cost.

### 4.1. Datasets and protocol

We use the four screening colonoscopy videos from the C3VD phantom dataset recorded by practicing gastroenterologists (Bobrow et al., 2023). Each sequence contains 4,700–5,500 frames and is accompanied by ground-truth camera poses, RGB video, and a watertight CAD mesh of the colon mold. We use per-frame monocular depth predictions from (Hardy et al., 2025) as supervision for all methods and resize images to $384 \times 384$.

To isolate the mapping/representation contribution, all methods are evaluated in a mapping-only setting with fixed ground-truth poses and identical depth supervision; this avoids conflating front-end tracking failures with reconstruction quality on long, fast-motion sequences. All methods are then evaluated under the same "every-other-frame" protocol, yielding an effective input rate of 15 fps. Every 8th processed frame is held out as a validation frame, with the remainder used for optimization; this ratio balances evaluation frequency against the number of frames available for map building. We run per-scene optimization over the entire sequence and evaluate online: at each processed frame, the current map is used to render all held-out frames seen so far. Additional low-level details are given in Appendix A.

### 4.2. Baselines

We compare against two 3D Gaussian mapping back-ends:
**EndoGSLAM.** EndoGSLAM is an endoscopy-specific 3D Gaussian SLAM system originally evaluated on short, robot-acquired C3VD sequences (Wang et al., 2024). For our long phantom sequences we disable tracking and pose refinement and feed ground-truth poses, so only the mapping component is active.
**MonoGS-style mapper.** The MonoGS baseline follows a standard 3D Gaussian mapping pipeline without colon-specific structure (Matsuki et al., 2024). It uses similar depth supervision and rendering losses as our method, but does not estimate a centerline, does not use tubular coordinates, and employs a conventional frame-based keyframe policy. For a controlled comparison, **Ours** and the **MonoGS-style mapper** share the same Gaussian mapping backbone (renderer, optimization loop, densification/pruning logic, opacity

Table 1: **Quantitative comparison on C3VD phantom sequences** (ground-truth poses). Values are mean $\pm$ standard deviation over four sequences. Higher is better for PSNR, SSIM, FPS; lower is better for CD. Per-sequence scores are in Appendix E.

| Method | PSNR ↑ | SSIM ↑ | FPS ↑ | CD ↓ | Points (M) |
|---|---|---|---|---|---|
| EndoGSLAM | $11.32 \pm 0.24$ | $0.346 \pm 0.025$ | $1.08 \pm 0.18$ | $6.61 \pm 1.35$ | $3.11 \pm 0.36$ |
| MonoGS | $11.26 \pm 0.66$ | $0.320 \pm 0.053$ | $8.20 \pm 0.67$ | $7.91 \pm 0.56$ | $0.59 \pm 0.07$ |
| Ours | $11.56 \pm 0.92$ | $0.335 \pm 0.057$ | $6.73 \pm 0.43$ | $5.73 \pm 0.58$ | $1.14 \pm 0.21$ |

reset policy, and iteration schedule); the only differences are the centerline/Bishop-frame module, arc-length keyframing, and the added colon-aware loss terms. **EndoGSLAM** is run with its standard mapping implementation with tracking and pose refinement disabled, ensuring the comparison reflects mapping behavior rather than front-end drift. Additional hyperparameters are listed in Appendix A.

All methods use exactly the same inputs (RGB, predicted depth, ground-truth poses) and run on the same NVIDIA RTX6000 GPU.

### 4.3. Metrics

**Reconstruction quality.** For each held-out frame $k$, we render $\hat{I}_k$ from its ground-truth pose and compute PSNR and SSIM with respect to the observed RGB image $I_k$. For geometry we compute a one-directional Chamfer distance (CD) from the reconstructed surface to the phantom mesh: we sample point clouds from both, compute nearest-neighbour distances from reconstruction points to mesh points, and average over points restricted to a fixed radial band around the centerline to ignore distant background.

**Runtime and memory.** Effective FPS is defined as FPS $= N/t$, where $N$ is the number of processed frames (including non-keyframes) and $t$ is the total wall-clock time for the sequence, including all components of each method. We also report the number of active Gaussians at the end of optimization.

**Coverage.** Using tubular coordinates $(s, r, \theta)$, we maintain online coverage statistics for each centerline segment and circumferential bin: (i) a scalar coverage score per segment (fraction of time the segment is within a viewing cone from the active camera) and (ii) a histogram of Gaussian counts over $\theta$ ("quadrants"). These metrics are updated in real time as $s$ grows and are later compared to a visibility oracle derived from the phantom mesh (Appendix E). We refer to these outputs as *geometric coverage*: they are visibility-based proxies derived from pose and geometry, and do not directly measure mucosal visualization quality under specularities, debris, blur, or occlusions behind folds.

### 4.4. Geometry–speed trade-off

Table 1 summarizes reconstruction quality, runtime, and model size, averaged over the four C3VD sequences; per-sequence results are given in Appendix E.

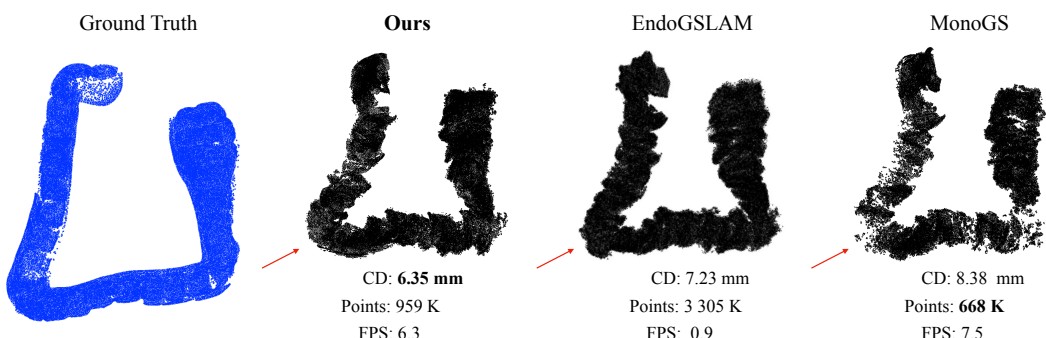

Figure 2: **Geometry–speed trade-off on a C3VD phantom sequence.** From left to right: ground-truth colon mesh and reconstructions from our centerline-aware mapper, EndoGSLAM, and a MonoGS-style mapper. Our method attains lower Chamfer distance than both baselines, uses roughly 3× fewer Gaussians than EndoGSLAM, and maintains near-MonoGS frame rates, yielding a thin tubular wall with fewer interior splats.

Our centerline-aware mapper matches or slightly improves PSNR relative to both baselines while achieving a Chamfer distance lower than both: we reach EndoGSLAM-level CD despite using fewer Gaussians, and improve CD by 2.2 mm on average over the MonoGS baseline. SSIM follows a similar pattern, with our method falling between EndoGSLAM and MonoGS; the difference across all three methods is small relative to the standard deviation, suggesting that structural similarity is not the primary axis of variation on this dataset. At the same time, our effective FPS is close to that of MonoGS and approximately 6× higher than EndoGSLAM, even though we maintain an online centerline, Bishop frame, and coverage counters. This trend holds consistently across all four sequences (Appendix E), supporting our claim that a thin geometric layer can recover much of the geometry that EndoGSLAM obtains from a heavier mapping stack while retaining MonoGS-like speed.

### 4.5. Qualitative geometry comparison

Figure 2 compares the reconstructed colon geometry on a representative C3VD sequence. We show the ground-truth phantom mesh alongside point clouds from our centerline-aware mapper, EndoGSLAM, and the MonoGS-style baseline, annotated with Chamfer distance (CD), number of active Gaussians, and effective FPS.

MonoGS achieves the highest FPS but allocates many Gaussians throughout the lumen, producing a thick, irregular tube and higher CD. EndoGSLAM concentrates points more tightly on the wall but at the cost of ∼ 3× more Gaussians and substantially lower FPS. Our method forms a thin, continuous tubular shell that more closely matches the phantom geometry while using fewer Gaussians than EndoGSLAM and running at near-MonoGS frame rates, agreeing with the quantitative trade-off in Table 1.

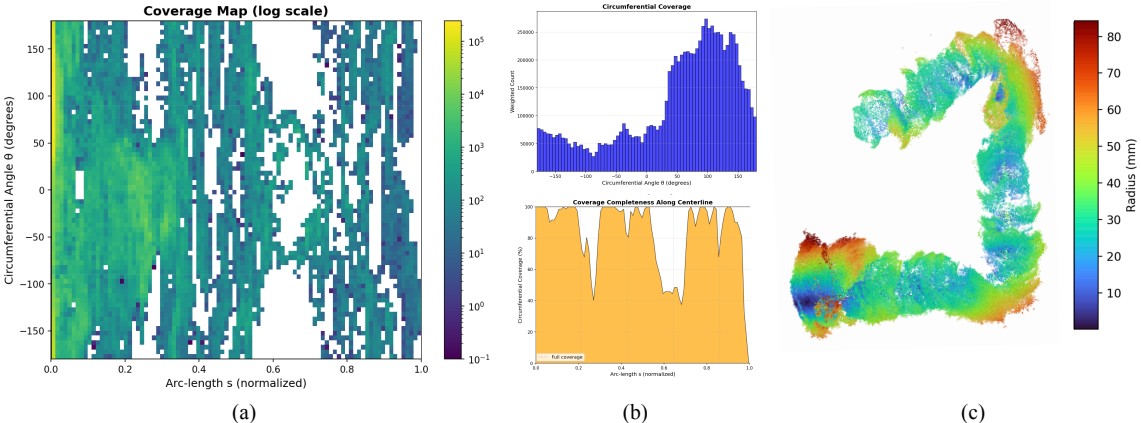

Figure 3: **Online coverage from tubular coordinates on a phantom sequence.**
(a) Unrolled coverage map in $(s, \theta)$: horizontal axis is arc length along the centerline, vertical axis is circumferential angle $\theta$, and color denotes our per-bin coverage score. Vertical lines mark anatomical segments of the phantom. (b) Segment-wise coverage summaries: top, bar plot of the fraction of surface area in each segment exceeding a minimum coverage threshold; bottom, coverage fraction as a function of $s$. (c) Circumferential balance: histogram of Gaussian counts over $\theta$, aggregated along the entire sequence (left) and for two example segments (right). Strong asymmetries indicate that the camera spent most of the time looking at one wall, suggesting that the opposite wall may warrant closer inspection.

### 4.6. Coverage in colon coordinates

A key advantage of expressing the map in tubular coordinates is that coverage can be computed online in the same representation. Figure 3 visualizes our coverage output for one phantom sequence.

Panel (a) shows the colon unrolled into $(s, \theta)$ with coverage encoded as a heatmap. Gaps or cold regions correspond to stretches that were rarely viewed with favourable distance and angle; in the phantom sequences these often occur around sharp bends and short withdrawal bursts. Panel (b) aggregates this into segment-wise summaries that can be inspected during or after a procedure, highlighting under-inspected regions. Panel (c) reports how Gaussians (and therefore map capacity) are distributed over circumferential angle. If a segment's Gaussians are heavily concentrated in one quadrant, it means the camera spent most of its time looking along that wall; the opposite wall may have received little attention even if overall time in that segment was adequate.

These coverage statistics are updated continuously as the centerline grows, with negligible additional cost: they reuse the same projections needed to render keyframes and require only per-segment, per-quadrant counters. In Appendix E we compare our online coverage scores to an oracle based on the phantom mesh and ground-truth poses and find good agreement, supporting their use as geometry-aware quality indicators rather than purely heuristic visualizations.

## 5. Conclusions and limitations

We have shown that a thin layer of colon-aware geometry on top of a standard 3D Gaussian mapper is enough to change the geometry–speed–utility trade-off. By organizing the map around an online centerline and tubular coordinates, we match or improve the Chamfer distance of an endoscopy-specific 3DGS baseline while running at frame rates close to a MonoGS-style mapper on long C3VD phantom sequences. The same representation yields unrolled colon views, segment-wise coverage summaries, and circumferential balance metrics with negligible additional computation, reusing projections already computed for rendering and requiring no new networks or changes to the underlying Gaussian rasterizer.

Our method has three main limitations. First, it assumes externally provided camera poses; in practice these would need to come from a robust SLAM or robotic tracking system, and our implementation does not correct global drift or miscalibration. Second, the pipeline is not yet real-time end-to-end at clinical frame rates and inherits the memory and compute demands of 3D Gaussian mapping. Third, our evaluation is restricted to silicone phantom data without annotated lesions, and the coverage metric is geometric: it captures pose-and-geometry-based visibility but does not account for fold occlusion, specularities, or debris, which are the conditions under which lesions are most commonly missed.

Despite these constraints, the coverage maps produced by our system are a natural substrate for real-time endoscopist feedback, post-procedure documentation, and quality auditing, connecting 3D reconstruction to the segment-level reporting that gastroenterologists already use in practice. Future work includes learning centerlines from image features, joint pose and centerline refinement, and validating coverage metrics against lesion-level endpoints in in vivo cohorts.

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

## Appendix A. Implementation details

**Backbone and optimization.**   Our implementation builds on a MonoGS-style 3D Gaussian mapping backbone with the same differentiable rasterizer and optimizer. Unless otherwise noted, we use the same hyperparameters for all sequences: Adam with a fixed learning rate, per-primitive opacity and covariance regularizers, and the depth and photometric losses described in Section 3.1. We process frames in temporal order and run a fixed number of gradient steps per keyframe; non-keyframes are used only to update the centerline, Bishop frame, and coverage counters.

**Hyperparameters.**   Table 2 summarizes the key centerline, keyframing, and prior parameters used throughout. Optimizer and batching details are given in Appendix B.

Table 2: Core hyperparameters, shared across all sequences.

| Parameter | Value |
|---|---|
| Keyframe spacing $\tau_{\text{KF}}$ (mm) | 20 |
| $\lambda_{\text{radial}}$, $\lambda_{\text{curv}}$ | 0.2, 0.02 |
| Robust penalty $\rho(\cdot)$ | Huber($\delta$) |
| $r_{\text{wall}}$ (mm) | 35 |
| Backbone spacing $d_{\text{min}}$ (mm) | 5 |
| Max bend angle $\theta_{\text{max}}$ (deg) | 120 |
| Spline degree | 3 (cubic) |
| Spline refit cadence $K_{\text{refit}}$ (points) | 5 |
| Centerline sampling step $\Delta s$ (mm) | 1 |

**Centerline update and thresholds.**   The backbone points used to define the centerline are updated online as in Section 3.2. We trigger a new backbone point when the camera center has moved at least $d_{\text{min}}$ along the trajectory, the incremental bend angle is below $\theta_{\text{max}}$, and the candidate is more than $d_{\text{loop}}$ from non-neighbor points (loop avoidance). We refit the centerline B-spline every $K_{\text{refit}}$ new backbone points and resample at arc-length step $\Delta s$; values are listed in Table 2. The refit is inexpensive compared to Gaussian optimization.

**Keyframing and continuous coverage updates.**   At each frame we update the arc-length accumulator $A_t$ and, when a keyframe is triggered, we (i) add the current frame to the optimization set, (ii) freeze the current coverage statistics for that frame, and (iii) reset $A_t$. Coverage counters are updated for every frame, not just keyframes: for the current camera pose we identify the closest centerline sample and increment coverage scores for bins within a small arc-length neighborhood of this sample and within a viewing cone of the camera ray. This allows coverage maps to be displayed continuously during the procedure without waiting for optimization to converge.

**Coverage view cone from intrinsics.**   The coverage viewing cone is defined by the camera intrinsics and image bounds: a candidate surface direction is considered observable if it projects inside the image under the intrinsics and lies within the distance and radial-band thresholds listed in Table 3. The effective angular extent is therefore determined by the camera field of view, not a learned parameter.

## Appendix B. Experimental protocol and evaluation details

**Hardware and measurement.** All methods are run on the same GPU with identical input streams (RGB, predicted depth, and fixed poses). Runtime is reported as effective FPS over full sequence wall-clock time.

**Optimization and batching.** We optimize per scene over the full sequence using only keyframes for back-end Gaussian updates. All optimizer settings (learning rate, schedules, and regularizer weights) are fixed across all sequences and are included in the released configuration.

## Appendix C. Chamfer distance and alignment

Although both the reconstruction and the phantom CAD mesh are expressed in millimeters, small residual calibration mismatches remain between coordinate frames. Before computing surface error we perform a rigid alignment with Iterative Closest Point (ICP), then compute a one-directional Chamfer distance from the reconstructed surface to the aligned phantom mesh.

**Rigid alignment.** For each sequence we extract a reconstructed surface point cloud $\mathcal{P}_{\text{traj}} = \{x_i\}$ from the Gaussian map and sample points $\mathcal{P}_{\text{obj}} = \{y_j\}$ from the phantom OBJ mesh. We run rigid ICP (point-to-plane) to find

$$T_{\text{ICP}} = \arg \min_{T \in SE(3)} \sum_i \text{dist}\big(Tx_i, \mathcal{P}_{\text{obj}}\big)^2,$$

and transform all reconstruction points by $T_{\text{ICP}}$. Camera poses are updated consistently so that rays still intersect the aligned phantom mesh.

**Visible phantom surface.** We restrict evaluation to the subset of the phantom surface actually observable given the camera trajectories. For every pixel $(u, v)$ in each frame with intrinsics $K$ and pose $T_t$ we: (i) back-project to a 3D ray using $K^{-1}$, (ii) transform into world coordinates using $T_t$, and (iii) compute the first intersection with the phantom mesh. Collecting all successful intersections gives $\mathcal{P}_{\text{hit}} = \{z_k\}$, the set of phantom points actually seen by the cameras. Rays are generated from the input intrinsics $K$ with a fixed pixel stride of 2, applied identically across all methods.

**One-sided surface error.** Our Chamfer distance is a one-sided RMS from the reconstructed surface to the visible phantom surface:

$$\text{CD} = \sqrt{\frac{1}{|\mathcal{P}_{\text{traj}}|} \sum_{x_i \in \mathcal{P}_{\text{traj}}} \min_{z_k \in \mathcal{P}_{\text{hit}}} \big\|x_i - z_k\big\|_2^2},$$

where $\mathcal{P}_{\text{traj}}$ denotes reconstruction points after applying $T_{\text{ICP}}$. We do not compute the reverse direction because parts of the phantom may never be visible; including them would penalize methods for failing to reconstruct unseen surfaces. All CD values are in millimeters using this definition with identical settings for all methods. Where evaluation is restricted to a radial band $r \in [r_{\min}, r_{\max}]$, the same band is applied consistently to both sampling and evaluation.

Table 3: Geometric coverage parameters used for Fig. 3 and all reported coverage summaries.

| Parameter | Value |
|---|---|
| View cone | camera frustum from $K$ |
| Max distance $d_{\max}$ (mm) | 100 |
| $r_{\text{wall}}$ (mm) | 35 |
| Arc-length neighborhood $\Delta s_{\text{cov}}$ (mm) | 20 |
| Radial band for wall evidence (mm) | $[25, 50]$ |
| $s$ binning | normalized by arc length |
| $\theta$ binning | 72 bins (5° each) |
| Per-frame update | +1 per visible Gaussian, $r \in [25, 50]$, depth $\leq d_{\max}$ |
| $\theta$ wrap-around | circular bins |

## Appendix D. Coverage oracle and metrics

To assess the accuracy of our online geometric coverage scores we construct a visibility oracle from the phantom mesh and ground-truth poses, using the same camera intrinsics $K$ and thus the same viewing angles.

**Coverage parameters.** Table 3 lists the thresholds and binning used to compute the unrolled $(s, \theta)$ coverage map and segment summaries.

**Online coverage scores.** At each frame we identify a local arc-length neighborhood around the camera's closest centerline coordinate and increment $(s, \theta)$ bins whose associated geometry lies within the distance and radial-band thresholds and projects inside the image under $K$. For comparison to the oracle we apply the normalization and thresholding rules in Table 3. These scores capture geometric coverage (pose- and geometry-based visibility) and do not directly measure mucosal visualization quality under blur, debris, specularities, or fold occlusions.

Table 4: Per-sequence results on C3VD phantom data (ground-truth poses).

| Method | Seq. | PSNR ↑ | SSIM ↑ | FPS ↑ | CD ↓ | # points |
|---|---|---|---|---|---|---|
| EndoGSLAM | v1 | 11.14 | 0.358 | 1.30 | 6.61 | 2 579 787 |
| | v2 | 11.35 | 0.368 | 0.94 | 7.68 | 3 306 894 |
| | v3 | 11.70 | 0.354 | 0.92 | 7.27 | 3 305 803 |
| | v4 | 11.08 | 0.302 | 1.15 | 9.74 | 3 261 161 |
| MonoGS | v1 | 12.13 | 0.398 | 9.11 | 7.76 | 624 033 |
| | v2 | 10.86 | 0.307 | 8.26 | 7.18 | 520 626 |
| | v3 | 11.40 | 0.292 | 7.58 | 8.38 | 668 002 |
| | v4 | 10.64 | 0.282 | 7.84 | 8.30 | 545 336 |
| Ours | v1 | 12.66 | 0.377 | 7.28 | 5.60 | 993 965 |
| | v2 | 10.93 | 0.327 | 6.24 | 4.99 | 1 430 093 |
| | v3 | 11.98 | 0.379 | 6.69 | 6.35 | 959 687 |
| | v4 | 10.69 | 0.259 | 6.70 | 5.96 | 1 163 906 |

## Appendix E. Per-sequence quantitative results

Figure 4 shows qualitative reconstructions across all four phantom sequences, complementing the single-sequence comparison in Figure 2. Table 4 reports per-sequence PSNR, SSIM, FPS, Chamfer distance, and number of active Gaussians for all methods.

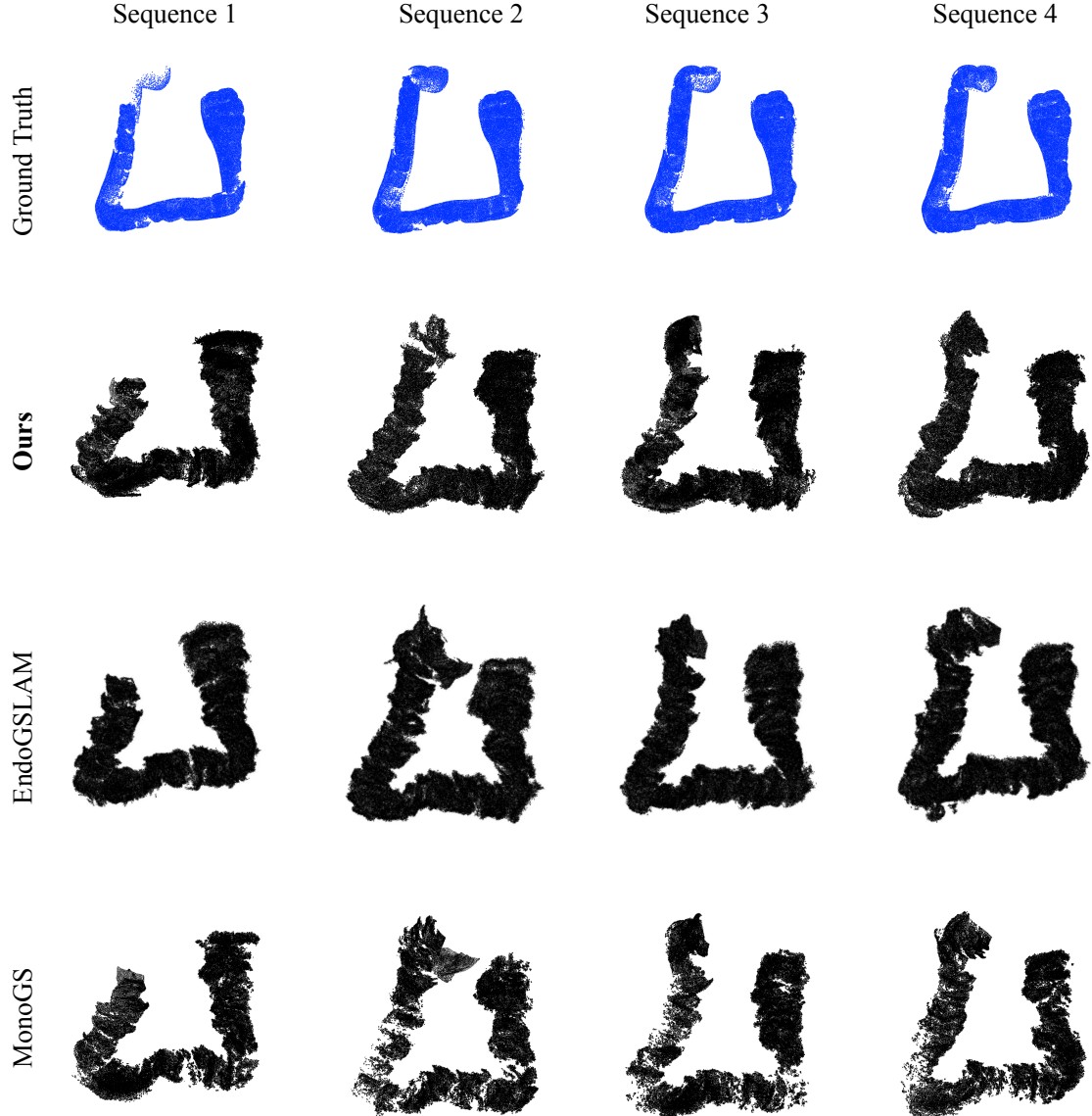

Figure 4: Qualitative reconstructions on all four C3VD phantom sequences. Gaussians are colored by radial distance from the centerline. Across sequences, Gaussians are consistently concentrated in a thin band around the colon wall with few interior points, illustrating the effect of the tubular prior and centerline-aware keyframing.

## Appendix F. Robustness to pose noise

To illustrate centerline behavior under pose noise we add synthetic perturbations to the ground-truth trajectories and recompute the centerline. We test two noise regimes: high-frequency local jitter, which the B-spline smoothing largely suppresses, and a slowly accumulating global bias, which displaces the centerline accordingly since our method does not attempt to correct systematic drift. Figure 5 shows representative results. In practice the centerline acts as a low-pass filter over camera motion and tolerates realistic tracker noise, but depends on a globally reasonable pose estimate from the upstream tracking system.

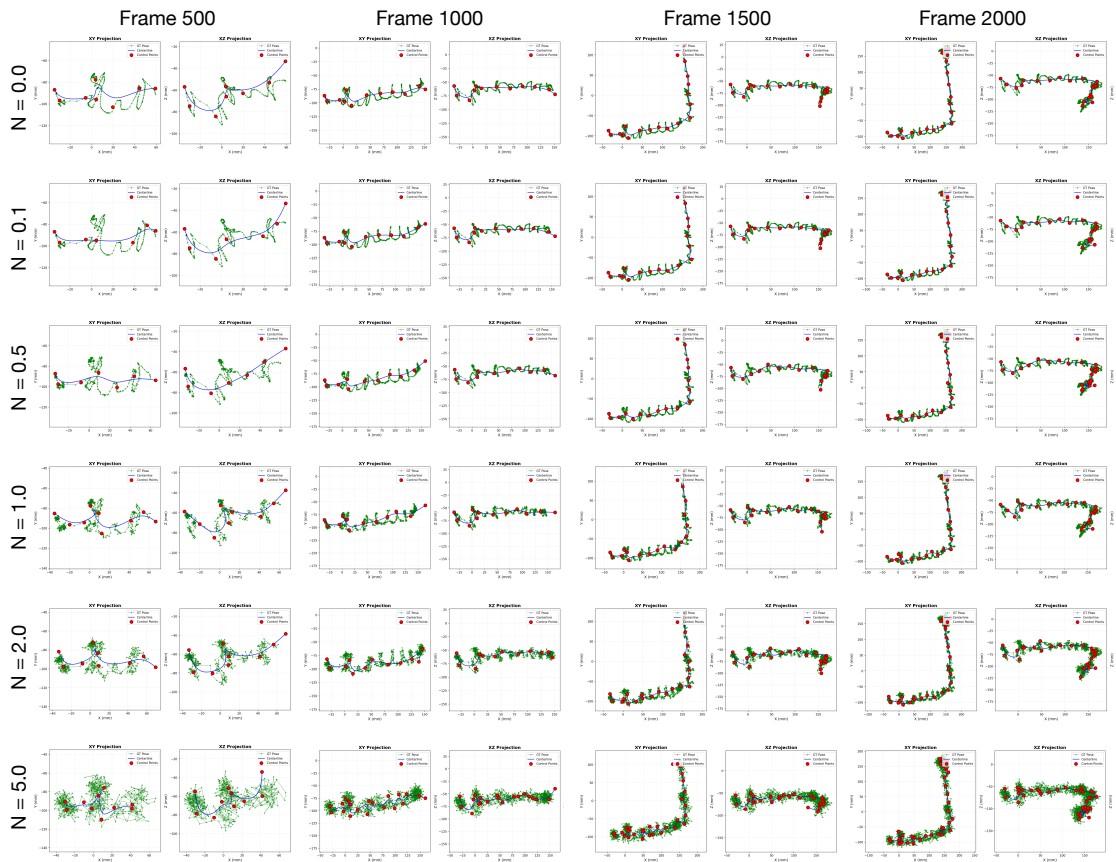

Figure 5: Effect of pose noise (N in mm) on the estimated centerline. Left: ground-truth trajectory and centerline. Middle: trajectory with high-frequency local jitter; the B-spline centerline remains smooth and close to the original. Right: trajectory with a slowly accumulating global bias; the centerline is displaced accordingly, as our method does not correct systematic drift.

## Appendix G. Bishop frame construction

For completeness we summarize the discrete Bishop-frame computation; the main text gives only the high-level description.

Given sampled centerline points $\{C_m\}_{m=0}^M$ with arc-length parameters $\{s_m\}$, we estimate unit tangents $T_m$ by centered finite differences

$$T_m = \frac{C_{m+1} - C_{m-1}}{\|C_{m+1} - C_{m-1}\|_2}, \qquad m = 1, \dots, M-1,$$

with one-sided differences at the endpoints. At $m = 0$ we choose an initial normal $N_{1,0}$ orthogonal to $T_0$ via

$$a_{\text{ref}} = \begin{cases} (0,1,0)^\top, & \text{if } |T_0 \cdot (0,0,1)^\top| > 0.9, \\ (0,0,1)^\top, & \text{otherwise,} \end{cases} \qquad N_{1,0} = \frac{T_0 \times a_{\text{ref}}}{\|T_0 \times a_{\text{ref}}\|_2}, \qquad N_{2,0} = \frac{T_0 \times N_{1,0}}{\|T_0 \times N_{1,0}\|_2}.$$

For $m = 1, \dots, M$ we propagate normals by discrete parallel transport. Let

$$a_m = T_{m-1} \times T_m, \qquad \theta_m = \text{atan2}(\|a_m\|_2, \, T_{m-1} \cdot T_m),$$

with unit axis $\hat{a}_m = a_m / \|a_m\|_2$ when $\|a_m\|_2 > \epsilon$ and $\hat{a}_m = T_m$ otherwise ($\epsilon = 10^{-6}$). We rotate $N_{1,m-1}$ by Rodrigues' formula

$$R(\hat{a}_m, \theta_m) = I + \sin\theta_m [\hat{a}_m]_\times + (1 - \cos\theta_m)[\hat{a}_m]_\times^2,$$

and set

$$N_{1,m} = R(\hat{a}_m, \theta_m)\, N_{1,m-1}, \qquad N_{2,m} = \frac{T_m \times N_{1,m}}{\|T_m \times N_{1,m}\|_2}.$$

**Sign-consistency check.** To prevent sign flips that would invert the $\theta$ axis, we check $\langle N_{1,m}, N_{1,m-1} \rangle$: if negative, we flip both normals $(N_{1,m}, N_{2,m}) \leftarrow (-N_{1,m}, -N_{2,m})$.

This discrete Bishop frame satisfies orthonormality up to numerical precision and minimizes twist along the curve, avoiding the instability of Frenet frames in nearly straight segments. In practice we re-orthogonalize by a single Gram–Schmidt step every few samples at negligible cost.

