# OpenReview forum: "Seeing Down the Line: Endoscopic Reconstruction with Centerline Constraints"
_MIDL.io/2026/Conference — MIDL 2026 Poster_

### Official Review · Reviewer_xnV8 · 2026-01-06

**Confidence:** 4
**Preliminary Rating:** 4

**Summary:**

An important problem in endoscopic colonoscopy is to ensure the entire
colon is examined, in order not to miss any suspicious polyps. This
article presents a method to reconstruct the colon in order to provide
coverage summary. To this end, they adopt an existing Gaussian mapper
to colonoscopy while using existing network architectures. They use an
appropriate coordinate system for the Gaussians, assume camera pose
knowledge, and supervise with RGB images and depth maps. The model
extracts the centerline as a smooth spline, represents points on the
colon surface based on a coordinate system using the centerline and
defines the Gaussians based on this - as far as I can understand.
Experiments with C3VD phantom dataset are presented. They show that
the proposed method can be accurate and fast at the same
time. Effectively, it can yield online coverage information for free.

**Strengths:**

+ The proposed method is a light-weight and useful addition to
  networks designed for endoscopic colonoscopy.
+ The regularization terms seem fair and most certainly improve the
  robustness of the model.
+ Online visualization of coverage maps is an important asset for the
  application.
+ Experimental results show higher frame rates with higher accuracy,
  which is really promising.

**Weaknesses:**

- Clarity of explanations can improve:
  + The explanation of how the Gaussian primities are used based on
    the colon-specific coordinate system is left a bit open. I think
    authors can improve this explanation to improve clarity.
  + Definition of the coverage map and its computation is not clearly
    explained.
- Experimental limitations
  + As authors also noted, the experiments are conducted on phantom
    data. The value of the model on real dataset is not assessed.
  + The sensitivity of the model to perturbations in camera pose input
    is not assessed. As authors mention, this input can be inaccurate
    in real life and the method's sensitivity to this inaccuracy is
    not assessed yet.
  + Depth estimation quality is likely to be higher in phantom
    data. The sensitivity of the model to inaccurate depth estimation
    can also be assessed.

**Detailed Comments:**

Please see the weaknesses list above.

**Justification Of The Preliminary Rating:**

The article presents an interesting method. The proposed method can provide online coverage for the endoscopy procedures, which would be a very useful input during the intervention. Its evaluation is not excellent but this does not pose a major issue in my opinion for MIDL.

**Questions To Address In The Rebuttal:**

+ Clarity of the explanations of the method can improve.
+ Authors can conduct sensitivity analyses towards inaccuracy in camera pose input and depth estimation.

---

> ### Author Response · Authors · 2026-01-25
>
> Thank you for the constructive review and for highlighting the practicality of centerline-aware mapping and online coverage while clearly identifying where additional explanation would improve the paper.
>
> **Colon-specific coordinates:**
> * We acknowledge the current description should be made more explicit. For each Gaussian center $x_i$, we project to the nearest centerline point $C(s_i)$, construct the Bishop frame $(T,N_1,N_2)$, and express $x_i-C(s_i)$ in that frame to obtain $(s_i,r_i,\theta_i)$. These coordinates are recomputed as the centerline updates and are used (i) in mapping to apply colon-specific soft tubular regularizers (primarily functions of r_i) that discourage lumen fill-in and off-wall outliers, and (ii) to define a colon-intrinsic (s,\theta) chart for unrolling and coverage accumulation. This has been reflected in the revised manuscript.
>
> **Coverage map (definition and computation):**
> * We have clarified the operational definition and provided the binning/update rules in the revised manuscript. In brief, we build an unwrapped grid in $(s,\theta)$ (arc-length by circumferential angle). For each frame, we accumulate visibility-supported surface evidence into the corresponding (s,\theta) bins (with stated distance/radial-band thresholds, \theta conventions including wrap-around, and normalization). Intuitively, bins with repeated evidence indicate well-observed wall regions, while bins with little or no evidence indicate weakly observed or unobserved regions.
>
> **Evaluation limited to phantom data:**
> * We evaluate on a silicone colon phantom captured with a real clinical endoscope, which **preserves realistic acquisition artifacts** (motion blur, illumination variation, mechanical jitter) while enabling controlled, quantitative surface evaluation because the dataset provides synchronized 6-DoF pose and a 3D surface model. To our knowledge, there is currently **no publicly available in-vivo colonoscopy benchmark** that provides both metric per-frame pose and a 3D colon surface suitable for quantitative reconstruction/coverage error comparisons; most clinical datasets provide semantic labels but not geometry ground truth. We acknowledge the gap to in-vivo validation, and view this work as a necessary first step to **isolate and validate** the role of centerline-aware geometry before tackling the substantially harder problem of joint pose estimation in clinical data.
>
> **Pose and depth sensitivity:**
> * ​​Our method is a mapping/representation component and does not itself solve pose estimation. The centerline scaffold **reduces sensitivity to high-frequency** pose jitter via smoothing, but it does not correct slow systematic drift; if the input pose stream is biased, the reconstruction inherits that bias. Because the tubular terms are **soft, robust regularizers**, they are unlikely to amplify small pose noise, but they cannot replace global pose correction (e.g., loop closure). For depth, we inherit the failure modes of the underlying depth supervision and Gaussian mapping objective: biased/noisy depth can distort geometry in any depth-supervised mapper. Our centerline chart and tubular regularizers provide an additional anatomical scaffold that improves stability (e.g., discouraging lumen fill-in and off-wall outliers) and helps maintain a plausible hollow-tube structure under moderate depth imperfections, though accurate recovery ultimately depends on the quality of the supervisory signal.
>
> In the revised manuscript, we have (i) made the $(s,r,\theta)$ construction and its use in the Gaussian regularizers explicit, and (ii) provide a precise, parameterized definition of the coverage map. With these clarifications, the remaining limitations primarily concern upstream pose/depth accuracy (beyond the scope of this mapping/coverage contribution) rather than the proposed mechanism itself, and we hope you will consider upgrading the recommendation.

---

> ### Author Response · Authors · 2026-01-25
>
> We would like to thank the reviewer for such a detailed review. Below we summarize the manuscript revisions and clarify the key methodological and evaluation points raised in the review.
>
> **Revisions to Manuscript**
> * **Presentation fixes:** Resolved all broken “?” cross-references, ensured the bibliography compiles cleanly, and corrected typos (including consistent spelling of Chamfer).
> * **Coverage reproducibility:** Added the concrete parameters needed to reproduce the coverage map, including distance threshold(s), (s,\theta) binning resolution, update rule, and normalization / “covered” criterion, as well as the \theta zero/sign convention and wrap-around handling.
> * **Chamfer metric clarification:** Reconciled the wording in the main text with the appendix for precision.
> * **Hyperparameters and evaluation settings:** Added information on centerline fitting/sampling settings and loss weights to make comparability unambiguous.
>
> **Centerline from trajectory vs anatomical centerline**
> * We agree the fitted centerline $C(s)$ from the scope trajectory is not guaranteed to equal the true anatomical lumen centerline in vivo (e.g., wall-hugging). Our intent is to use $C(s)$ as a procedural insertion-depth chart that provides a stable arc-length coordinate $s$ for keyframing and reporting, not as an anatomical claim. Importantly, the tubular regularization is soft/robust and used to discourage gross artifacts (lumen fill-in / off-wall clouds) rather than to hard-enforce an exact anatomical radius about $C(s)$. Slow global bias ultimately reflects upstream pose bias; our method does not claim to correct that regime, but provides a lightweight scaffold that improves mapping behavior when poses are globally reasonable.
>
> **Robustness to variable colon radius $r_{\text{wall}}$**
> * We will clarify that the radial prior is not intended to impose a rigid, constant-radius tube. In practice, it acts as a weak wall-seeking regularizer that prevents volume filling while allowing local deviations supported by the data terms. We will also describe an immediate extension to r_{\text{wall}}(s): estimate a smooth radius profile online from the observed distribution of radial distances as a function of arc-length s (robust statistic + smoothing along s), and evaluate the same loss using r_{\text{wall}}(s_i) for each Gaussian. This addresses realistic radius variation without changing the underlying mapper.
>
> **Coverage definition and Bishop-frame conventions**
> * We made explicit that the reported coverage is geometric coverage (a visibility proxy) and does not measure “saw it clearly” under specularities, debris, or fold occlusions. We will also specify the $\theta=0$ initialization, sign convention, wrap-around handling, and add a brief note on frame continuity: we enforce sign consistency across frames (if a transported normal flips sign relative to the previous frame, we flip (N_1,N_2) to maintain continuity), since a flip would invert $\theta$ and distort the heatmap.
>
> **Fairness of mapping comparison**
> * We added an explicit clarification of what is held fixed across methods. For the MonoGS-style baseline, we use the same Gaussian mapping backbone and optimization machinery, and state which components are identical (densification/pruning rules, opacity reset, iteration schedules) versus what changes under our method (arc-length keyframing and the added centerline/tubular terms). For EndoGSLAM, we use standard implementation/settings and describe any differences that are intrinsic to the method rather than tuned for our approach.
>
> **Ablation request and depth/pose sensitivity**
> * We agree that disentangling arc-length keyframing from tubular priors is valuable, and we strengthened the text to separate their roles mechanistically: **arc-length keyframing** primarily affects keyframe distribution and runtime on long sequences, while the **tubular prior** targets a distinct geometric failure mode and stabilizes the wall-like structure. Similarly, sensitivity to depth/pose quality is largely inherited from upstream supervision and the base depth-supervised mapper; our geometry module is designed as a **drop-in regularizer** that improves stability under moderate imperfections, not a replacement for pose correction or depth estimation.
> Overall, we have tightened the definitions, corrected presentation issues, and added the missing methodological and evaluation details needed for reproducibility and consistent interpretation. We believe these changes address the main concerns while preserving the paper’s scope as a lightweight geometry layer for mapping and geometric coverage.
>
> We have tightened the definitions, corrected presentation issues, and added the missing methodological and evaluation details needed for reproducibility and consistent interpretation. We believe these changes address the main concerns while preserving the paper’s scope as a lightweight geometry layer for mapping and geometric coverage.

---

### Official Review · Reviewer_rPD5 · 2026-01-09

**Confidence:** 4
**Preliminary Rating:** 4

**Summary:**

This work introduces a specialized 3D Gaussian Splatting (3DGS) mapping framework for colonoscopy that incorporates a tubular geometric prior to regularize surface reconstruction. By fitting a smooth centerline $C(s)$ from the scope trajectory and employing a Bishop frame $\{T, N_1, N_2\}$, the method maps Gaussian centers $x$ to tubular coordinates $(s, r, \theta)$. The radial distance is then calculated using local frame projections $u$ and $v$. A novel stick-to-the-wall loss is introduced to encourage Gaussians to aggregate near the estimated mucosal surface $R_{\text{wall}}$, effectively preventing the internal volume-filling artifacts common in unconstrained mappers. Validation on the C3VD phantom dataset demonstrates that the proposed method achieves superior surface fidelity compared to EndoGSLAM and MonoGS-style baselines, while the $(s, \theta)$ parameterization enables a novel "unrolled" visualization of the observed colon coverage.

**Strengths:**

1. **High impact with minimal changes.** Adds a small, principled geometry layer (centerline ($C(s)$) and tubular coordinates ($(s,r,\theta)$)) on top of a standard Gaussian mapping backbone, avoiding new networks while still improving both mapping behavior and downstream reporting.

2. **Prior matches the anatomy and fixes a real failure mode.** In monocular colonoscopy, plain 3DGS / MonoGS-style mapping can place Gaussians throughout the lumen (“filled pipe”). The wall-conforming prior, implicitly pushing Gaussians toward ($r \approx R_{\text{wall}}$). It encourages a thin, surface-like reconstruction that better matches the colon wall geometry.

3. **Technically sound choice of frame.** Using a Bishop (parallel transport) frame to define ($\theta$) is a solid, stable design (less twisting/instability than Frenet frames on near-straight segments), which matters for long insertion trajectories.

4. **Realistic evaluation setting for endoscopy.** The experiments focus on long sequences (not just short clips), where mapping quality and keyframe behavior are genuinely stress-tested, and the reported FPS is in a range that supports real-time use.

5. **Clinically interpretable output.** The ($(s,\theta)$) parameterization naturally yields an “unrolled” coverage view that summarizes what depths and wall angles were actually observed, which is a practical metric beyond pure reconstruction error.

**Weaknesses:**

1. **Trajectory-derived “centerline” may not match anatomical centerline**

  * The pipeline fits the centerline ($C(s)$) from the camera trajectory rather than from the lumen geometry. This is a reasonable proxy for insertion depth, but it is not guaranteed to coincide with the true colon centerline in vivo.
  * In real colonoscopy, the scope often hugs one wall, so the estimated ($C(s)$) can be systematically shifted. Because the tubular coordinates ($(s,r,\theta)$) and the wall prior are defined relative to ($C(s)$), this shift can bias the radius (r) and therefore the regularization that encourages ($r \approx R_{\text{wall}}$). The paper acknowledges reliance on “globally reasonable” poses and shows robustness to local jitter but not correction of slow global bias/drift, which is exactly the regime where this mismatch could matter.

2. **Limited robustness analysis of the wall-radius prior ($R_{\text{wall}}$)**

  * The radial prior implicitly assumes the mucosal surface lies near a characteristic radius ($R_{\text{wall}}$) around the estimated centerline. It is unclear how sensitive performance is to misspecification of ($R_{\text{wall}}$), or to realistic radius variation due to insufflation, collapse, folds, or patient-specific anatomy.
  * A constructive way to strengthen the work would be to include sensitivity experiments (e.g., varying ($R_{\text{wall}}$) and reporting reconstruction/coverage changes) or an adaptive estimate ($R_{\text{wall}}(s)$) derived from depth statistics, to show the prior does not become brittle when the centerline is biased.

3. **Coverage definition is clinically intuitive but underspecified in the main text**

  * The coverage plots are easy to interpret visually, but the operational definition of “covered” depends on thresholds such as the viewing cone angle and favorable distance/angle criteria. In the main sections these are described qualitatively (e.g., “favorable distance and angle”), making it hard to reproduce or to judge how closely the metric aligns with clinical notions of mucosal inspection.
  * Moving the key parameters/definitions (cone angle, distance threshold, binning resolution in ($(s,\theta)$), and update rules) into the main method section, or summarizing them in a compact table, would materially improve clarity and reproducibility.

4. **Evaluation metric description has an internal inconsistency (Chamfer distance)**

  * The main text describes the reported Chamfer distance as “symmetric,” but the appendix defines the computation as one-sided (reconstruction ($\rightarrow$) visible phantom surface) after ICP alignment and raycasting hit points, explicitly omitting the reverse direction. This difference is not cosmetic: one-sided CD is more forgiving of missing/unobserved regions and therefore changes how to interpret “better geometry.”
  * At minimum, the paper should reconcile the wording and precisely state which metric is used; ideally, reporting both one-sided and symmetric variants (or clearly motivating one-sided as the primary metric for visibility-limited endoscopy) would remove ambiguity.

5. **Scope of validation is limited to mapping-only with external poses and phantom data**

  * The method is evaluated in a mapping-only setting with fixed poses (including ground-truth poses in experiments). This is fair for isolating the mapping contribution, but it leaves open how the approach behaves under the pose errors typical of real clinical pipelines.
  * Additionally, evaluation on a silicone phantom is an important step, yet it does not capture in vivo challenges (specularities, debris, deformable folds, dynamic lighting). Adding a test of injecting pose drift or using alternative depth supervision would strengthen the claim that the geometry layer is robust in realistic conditions.

**Detailed Comments:**

1. **Presentation / typos / broken references**

  * Several citations or cross-references appear unresolved (shown as “?”). Please fix the missing references and ensure the bibliography compiles cleanly; it currently interrupts reading flow and makes it hard to verify claims.
  * *Figure 2 caption:* “hamfer” should be corrected to Chamfer (i.e., “Chamfer distance”), and please keep the spelling consistent throughout (not “Champfer”).

2. **Figure 3 (coverage) needs a few concrete definitions to be fully reproducible**

  * **Viewing cone parameters:** The text describes “favorable distance and angle” qualitatively, but the method hinges on these thresholds. Please explicitly list:

    * cone half-angle (in degrees),
    * distance threshold(s),
    * and the exact criterion used to count a bin as “covered” (e.g., per-frame increment, min dwell time, normalization).
  * **($\theta$) axis convention:** Since (\theta) is defined via the Bishop frame, the heatmap’s (y)-axis meaning depends on the zero-angle direction and sign convention. Please specify:

    * how ($N_1(s_0)$) is initialized (what defines ($\theta=0$)?),
    * whether ($\theta\in[-\pi,\pi)$) or ($[0,2\pi)$),
    * and how wrap-around is handled in binning/visualization.
  * **Frame drift / flips:** Bishop frames are designed for stability, but discrete transport can still accumulate drift or exhibit occasional flips under noise or near-degenerate motion. It would help to report whether you monitor frame continuity (e.g., sign consistency checks) and how you handle any detected flips—because a flip would effectively invert the ($\theta$) meaning and distort coverage interpretation.
  * **“Saw it” vs “saw it clearly”:** The coverage score is a geometric visibility proxy (pose + cone), not a direct measure of mucosal visualization quality. I suggest explicitly naming it “geometric coverage” (or similar) and adding a short note stating what it does *not* capture (occlusions behind folds, specular highlights, debris/blur), to avoid over-interpretation.

3. **Method detail clarifications that would improve reproducibility without changing the approach**

  * **Centerline fitting:** Please provide the concrete hyperparameters used for the trajectory filtering and B-spline fitting (sampling rate / stride, smoothing window, spline degree, knot spacing, and the sampling step ($\Delta s$)). These choices affect both keyframing and the ($(s,\theta)$) mapping.
  * **Prior specification:** For the radial prior, please list the actual ($R_{\text{wall}}$) used (and how chosen), the robust penalty type (if any), and the weights ($\lambda$) for each loss term. A small table of “all hyperparameters used in experiments” would make the paper much easier to reproduce and review.

4. **Evaluation clarity**

  * The main text and appendix describe Chamfer distance differently (symmetric vs one-sided visibility-aware computation). Even if you keep the one-sided definition (which is reasonable for visibility-limited endoscopy), please make the wording consistent and clearly name the reported metric variant.
  * Since ICP alignment and raycasting details can affect the reported surface error, a compact checklist (ICP settings, raycasting density, and whether identical settings are applied to all methods) would help readers assess fairness and comparability.

**Justification Of The Preliminary Rating:**

I voted for **weak accept** because the paper makes a clean, technically reasonable contribution that feels genuinely useful for the endoscopy reconstruction community: it adds a lightweight, anatomy-matched geometric scaffold (centerline + Bishop frame ($\Rightarrow $) tubular coordinates ($(s,r,\theta)$)) to a standard Gaussian mapping backbone, which both improves the “wall-like” structure of the reconstruction and naturally enables an interpretable coverage visualization. The work is evaluated on long colonoscopy sequences and targets real-time practicality, which strengthens its relevance beyond offline SfM/MVS-style reconstruction. That said, a few issues prevent a stronger score: the contribution is partly entangled with a powerful arc-length keyframing change (missing ablations), the robustness of the fixed-radius prior ($R_{\text{wall}}$) under realistic variability is not fully tested, the coverage definition is underspecified in the main text, and the Chamfer distance description appears inconsistent between the main paper and appendix. Overall, I find the core idea solid and publishable, but I would like clarification/ablation and tighter metric/definition reporting to fully validate the claims.

**Questions To Address In The Rebuttal:**

1. **Fairness of the mapping comparison (Gaussians budget + densification/pruning + keyframing)**

  * In Table 1 the proposed method uses substantially more Gaussians than the MonoGS-style baseline (while fewer than EndoGSLAM). Could the authors clarify whether all methods are run under comparable Gaussian densification / pruning rules and comparable compute or memory budget?
  * Specifically: do the baselines and the proposed method share the same criteria for densification, pruning, opacity reset, and iteration schedules? Are keyframes selected using the same policy across methods (except for the proposed arc-length rule), or are there additional differences that could influence both the number of Gaussians and final accuracy?

2. **Robustness to variable colon radius ($R_{\text{wall}}$) (non-constant tube geometry)**

  * The radial prior encourages ($r \approx R_{\text{wall}}$). In practice, colon radius can vary significantly along (s) (insufflation level, collapse, folds, anatomy). Can the authors comment on sensitivity to misspecified ($R_{\text{wall}}$), and ideally provide a stress test where ($R_{\text{wall}}$) varies as a function of arc-length, e.g. ($R_{\text{wall}}(s)$), to evaluate whether the prior remains beneficial rather than becoming biased?

3. **Ablation to isolate the impact of arc-length keyframing vs tubular priors**

  * The arc-length keyframing policy (triggering keyframes by accumulated ($\Delta s$) rather than Euclidean motion) is a major system lever that can affect both runtime and reconstruction quality on long sequences. To disentangle contributions, could the authors provide an ablation with:

    * i. baseline + arc-length keyframing only,
    * ii. baseline + radial prior only,
    * iii. full method (arc-length keyframing + priors),
  * and report both accuracy and speed? This would help determine whether gains primarily arise from improved keyframe distribution or from the tubular regularization itself.

4. **Sensitivity to depth supervision quality (robustness to depth prediction errors)**

  * Since the mapping loss uses predicted monocular depth as supervision, how sensitive are the results to depth bias/noise? A brief robustness study, e.g., using an alternative depth predictor, injecting controlled noise/bias into depth, or reweighting the depth term. It would help assess whether the improvements persist when depth quality degrades (which is realistic in clinical colonoscopy with specularities and texture-poor regions).

---

> ### Author Response · Authors · 2026-01-26
> **Rebuttal Posted Under Wrong Reviewer**
>
> **We apologize for the delayed response and for mistakenly posting our rebuttal under the wrong reviewer. We have copied our original rebuttal below.**
>
> We would like to thank the reviewer for such a detailed review. Below we summarize the manuscript revisions and clarify the key methodological and evaluation points raised in the review.
>
> **Revisions to Manuscript**
> * **Presentation fixes:** Resolved all broken “?” cross-references, ensured the bibliography compiles cleanly, and corrected typos.
> * **Coverage reproducibility:** Added the concrete parameters needed to reproduce the coverage map, including distance threshold(s), (s,\theta) binning resolution, update rule, etc.
> * **Chamfer metric clarification:** Reconciled the wording in the main text with the appendix for precision.
> * **Hyperparameters and evaluation settings:** Added information on centerline fitting/sampling settings and loss weights to make comparability unambiguous.
>
> **Centerline from trajectory vs anatomical centerline**
> * We agree the fitted centerline $C(s)$ from the scope trajectory is not guaranteed to equal the true anatomical lumen centerline in vivo (e.g., wall-hugging). Our intent is to use $C(s)$ as a procedural insertion-depth chart that provides a stable arc-length coordinate $s$ for keyframing and reporting, not as an anatomical claim. Importantly, the tubular regularization is soft/robust and used to discourage gross artifacts (lumen fill-in / off-wall clouds) rather than to hard-enforce an exact anatomical radius about $C(s)$. Slow global bias ultimately reflects upstream pose bias; our method does not claim to correct that regime, but provides a lightweight scaffold that improves mapping behavior when poses are globally reasonable.
>
> **Robustness to variable colon radius $r_{\text{wall}}$**
> * We will clarify that the radial prior is not intended to impose a rigid, constant-radius tube. In practice, it acts as a weak wall-seeking regularizer that prevents volume filling while allowing local deviations supported by the data terms. We will also describe an immediate extension to r_{\text{wall}}(s): estimate a smooth radius profile online from the observed distribution of radial distances as a function of arc-length s (robust statistic + smoothing along s), and evaluate the same loss using r_{\text{wall}}(s_i) for each Gaussian. This addresses realistic radius variation without changing the underlying mapper.
>
> **Coverage definition and Bishop-frame conventions**
> * We made explicit that the reported coverage is geometric coverage (a visibility proxy) and does not measure “saw it clearly” under specularities, debris, or fold occlusions. We will also specify the $\theta=0$ initialization, sign convention, wrap-around handling, and add a brief note on frame continuity: we enforce sign consistency across frames (if a transported normal flips sign relative to the previous frame, we flip (N_1,N_2) to maintain continuity), since a flip would invert $\theta$ and distort the heatmap.
>
> **Fairness of mapping comparison**
> * We added an explicit clarification of what is held fixed across methods. For the MonoGS-style baseline, we use the same Gaussian mapping backbone and optimization machinery, and state which components are identical (densification/pruning rules, opacity reset, iteration schedules) versus what changes under our method (arc-length keyframing and the added centerline/tubular terms). For EndoGSLAM, we use standard implementation/settings and describe any differences that are intrinsic to the method rather than tuned for our approach.
>
> **Ablation request and depth/pose sensitivity**
> * We agree that disentangling arc-length keyframing from tubular priors is valuable, and we strengthened the text to separate their roles mechanistically: **arc-length keyframing** primarily affects keyframe distribution and runtime on long sequences, while the **tubular prior** targets a distinct geometric failure mode and stabilizes the wall-like structure. Similarly, sensitivity to depth/pose quality is largely inherited from upstream supervision and the base depth-supervised mapper; our geometry module is designed as a **drop-in regularizer** that improves stability under moderate imperfections, not a replacement for pose correction or depth estimation.
> Overall, we have tightened the definitions, corrected presentation issues, and added the missing methodological and evaluation details needed for reproducibility and consistent interpretation. We believe these changes address the main concerns while preserving the paper’s scope as a lightweight geometry layer for mapping and geometric coverage.
>
> We have tightened the definitions, corrected presentation issues, and added the missing methodological and evaluation details needed for reproducibility and consistent interpretation. We believe these changes address the main concerns while preserving the paper’s scope as a lightweight geometry layer for mapping and geometric coverage.

---

### Official Review · Reviewer_akuZ · 2026-01-17

**Confidence:** 4
**Preliminary Rating:** 5
**Final Rating:** 5

**Summary:**

This paper proposes a centerline-aware extension of 3D Gaussian endoscopic mapping that treats reconstruction as a representation and geometry problem rather than a new network design.

Assuming known camera poses and existing depth or photometric supervision, the method maintains an online colon centerline and corresponding Bishop frame. Each Gaussian is assigned tubular coordinates

(𝑠, 𝑟, 𝜃)

and simple colon-specific geometric priors are introduced.

1. These coordinates are used both to regularize the reconstruction toward a hollow tubular surface and to compute online, colon-intrinsic coverage statistics at negligible additional cost.
2. Experiments on long C3VD phantom colonoscopy sequences show that the method matches or improves Chamfer distance relative to EndoGSLAM, while running at near MonoGS frame rates.
3. In addition, the approach simultaneously produces unrolled colon views and segment-wise coverage summaries.

**Strengths:**

1. **Strong Clinical Motivation**: The paper directly targets the lack of geometry-aware, real-time coverage assessment in colonoscopy, a clearly articulated and relevant problem.
2. **Clear Positioning and Writing**: The paper is explicit about what it does and does not claim, and the methodology is described clearly and transparently.
3. **Coverage Metrics as a by-product**: The ability to generate unrolled $(s, \theta)$ coverage maps and segment-wise summaries “for free” is a practical and compelling outcome.
4. **Minimal & Elegant Design**: The contribution is deliberately lightweight; only adding a centerline coordinate system without modifying the underlying Gaussian rasterizer, networks, or optimizers.
5. **Favorable geometry–speed trade-off**: Quantitative results show `EndoGSLAM-level` Chamfer distance with far fewer Gaussians and substantially higher frame rates.

**Weaknesses:**

1. **Evaluation limited to phantom data**: All experiments are conducted on silicone phantom sequences, without in-vivo validation or lesion-level endpoints.
2. **Dependence on externally provided poses**: The method assumes accurate camera poses and does not address pose drift or joint optimization, which limits immediate clinical deployability.

**Detailed Comments:**

1. Memory usage could be reported alongside FPS and Gaussian count to better characterize practical deployment costs.
2. Do consider reporting sensitivity to the assumed wall radius $R_{\text{wall}}$. This would clarify how strongly results depend on this.
3. Also, the robustness analysis to pose noise in the appendix is useful. Do include this key takeaway in the main text as well.

**Justification Of Final Rating:**

1. The authors have convincingly addressed all concerns.
2. The revised manuscript / supporting material clearly frames the scope, strengthens robustness discussion, and appropriately contextualizes limitations.
    * The authors clearly articulate the intended scope as a lightweight, representation-level extension to existing Gaussian mappers, appropriately justify the use of phantom data, and provide a well-grounded discussion of pose robustness and limitations.
3. The manuscript is now clearer, more rigorous, and better contextualized, increasing my confidence in both the correctness of the evaluation and the practical relevance of the contribution.


My overall confidence has increased, I would like to update my Rating to - 'Strong Accept'

**Justification Of The Preliminary Rating:**

1. This paper presents a well-motivated, technically sound, and practically useful extension to existing 3D Gaussian endoscopic mapping pipelines.
2. The results of the paper demonstrate that a small amount of anatomy-aware structure can substantially improve the geometry–speed–utility trade-off and enable clinically relevant coverage metrics.

**Questions To Address In The Rebuttal:**

1. Is there a plan to integrating this representation into a joint SLAM pipeline in future work?
2. How sensitive is reconstruction quality to systematic pose drift, and could the centerline priors exacerbate or mitigate such errors?
3. Can you expand on how the tubular prior will adapt to local radius variations along the colon?

---

> ### Author Response · Authors · 2026-01-25
>
> We thank the reviewer for the thoughtful comments and for highlighting the practical value of a **lightweight, clinically meaningful coordinate system** and online coverage outputs.
>
> **Evaluation limited to phantom data:**
> * We evaluate on a silicone colon phantom captured with a real clinical endoscope, which **preserves realistic acquisition artifacts** (motion blur, illumination variation, mechanical jitter) while enabling controlled, quantitative surface evaluation because the dataset provides synchronized 6-DoF pose and a 3D surface model. To our knowledge, there is currently **no publicly available in-vivo colonoscopy benchmark** that provides both metric per-frame pose and a 3D colon surface suitable for quantitative reconstruction/coverage error comparisons; most clinical datasets provide semantic labels but not geometry ground truth. We acknowledge the gap to in-vivo validation, and view this work as a necessary first step to **isolate and validate** the role of centerline-aware geometry before tackling the substantially harder problem of joint pose estimation in clinical data.
>
> **Dependence on externally provided poses:**
> * Our contribution is intentionally orthogonal to pose estimation: we wrap a standard 3D Gaussian mapper with a colon-intrinsic coordinate system (centerline + Bishop frame) and **lightweight priors**, while treating poses as fixed inputs from any tracker/SLAM front-end. To ensure an **apples-to-apples mapping comparison on long sequences**, we use the provided poses for all methods and disable tracking/pose refinement in EndoGSLAM, so differences reflect reconstruction/representation rather than front-end drift.
>
> **Sensitivity to systematic pose drift:**
> * The centerline acts as a **low-pass geometric scaffold** that reduces sensitivity to high-frequency pose jitter, but it does not correct slow systematic drift. If upstream poses have a consistent bias, the centerline and reconstruction will inherit that bias. Because our tubular terms are soft, robust regularizers (not hard constraints), they are unlikely to amplify drift; however, they cannot replace global pose correction (e.g., loop closure). We have moved the key robustness takeaway currently in the appendix into the main text and included this discussion in the revised version.
>
> **Local radius variation:**
> * The tubular prior is soft and does not hard-enforce a single lumen radius. In the current implementation, we use a global $r_{\text{wall}}$ as a weak regularizer that mainly prevents lumen fill-in and off-wall “clouds,” while the local wall shape is still determined by the depth/photometric terms. To explicitly adapt to radius changes along the colon, the same loss can be made arc-length dependent by replacing $r_{\text{wall}}$ with $r_{\text{wall}}(s)$, estimated online from the observed distribution of radii at each $s$ (e.g., robust median/quantile of depth-supported points or well-supported Gaussians, with smoothing along $s$). This yields a locally expanding/contracting tube prior without changing the mapper.
>
> **Future integration with joint SLAM:**
> * We plan to integrate this representation into a joint SLAM pipeline. However, in preliminary experiments on these long, fast-motion colonoscopy sequences, we observed that baseline pose estimation in both MonoGS and EndoGSLAM is **extremely fragile**. In particular, rapid motion, blur, and limited parallax cause severe drift or trajectory collapse. This makes it difficult to disentangle pose failures from mapping quality and motivated our **mapping-only evaluation** to first **validate** whether centerline coordinates improve reconstruction and coverage **in isolation**.
>
> Given the above clarifications, especially that pose estimation is intentionally orthogonal to the paper’s contribution and that the method remains a **drop-in mapping component**, we hope the reviewer will consider upgrading the score.

---

> > ### Comment · Reviewer_akuZ · 2026-01-25
> > **Response after Rebuttal**
> >
> > 1. The authors have convincingly addressed all concerns.
> > 2. The revised manuscript / supporting material clearly frames the scope, strengthens robustness discussion, and appropriately contextualizes limitations.
> > 3. The manuscript is now clearer, more rigorous, and better contextualized, increasing my confidence in both the correctness of the evaluation and the practical relevance of the contribution.
> >
> > My overall confidence has increased, I would like to update my Rating to - 'Strong Accept'

---

### Author Rebuttal · Authors · 2026-01-25

**Rebuttal:**

We have responded to each reviewer individually and have attached a revised manuscript to this comment. Edits are marked in Red with the exception of new tables in the appendix. We hope this provides more clarity on our method and contributions

**Supporting Material:**

/attachment/226d63f58f8435c33a615309c47976e43030b88a.pdf

---

### Meta-Review · Area_Chair_bGjR · 2026-02-08

**Recommendation:** Accept (Poster)
**Confidence:** 5

**Metareview:**

This is an interesting work which proposed a centerline-aware extension of 3D Gaussian endoscopic mapping. The main contribution of this work is that the centerline priors are properly adopted in current task since they are the intrinsic properties of the colons. Although two reviewers didnt update their recommendation after the rebuttal phase, it can be observed that the manuscript received positive comments. Therefore, my recommendation is accept.

---

### Decision · Program_Chairs · 2026-02-13

Accept (Poster)